

# Seasonal snow cover indicators in coastal Greenland from in-situ observations, a climate model and reanalysis

Jorrit van der Schot[1,3], Jakob Abermann[1,3], Tiago Silva[1], Kerstin Rasmussen[1,3], Michael Winkler[2], Kirsty Langley[4], Wolfgang Schöner[1,3]

[1]Department of Geography and Regional Science, University of Graz, Graz, 8010 Graz, Austria
[2]GeoSphere Austria, A-6020 Innsbruck, Austria
[3]Austrian Polar Research Institute, Wien, 1030 Wien, Austria
[4] Asiaq - Greenland Survey, Nuuk, Nuuk 3900, Greenland

*Correspondence to*: Jorrit van der Schot (jorrit.van-der-schot@uni-graz.at)

**Abstract.** Seasonal snow cover has important climatic and ecological implications for the ice-free regions of coastal Greenland. Here we present, for the first time, a dataset of quality-controlled snow depth measurements from nine locations in coastal Greenland with varying periods between 1997 and 2021. Using a simple modelling approach (Δsnow) we estimate snow water equivalent values solely based on the daily time series of snow depth. Snow pit measurements from two locations enable us to evaluate the Δsnow model. As there is very little in-situ data available for Greenland, we then test the performance

of the regional atmospheric climate model (RACMO2.3p2, 5.5 km spatial resolution) and reanalysis product (CARRA, 2.5 km spatial resolution) at the nine locations with snow observations. Using the combined information from all three data sources, we study spatio-temporal characteristics of the seasonal snow cover in coastal Greenland by the example of six ecologically relevant snow indicators (maximum snow water equivalent, melt onset, melt duration, snow cover duration, snow cover onset, snow cover end). In particular, we evaluate the ability of RACMO2.3p2 and CARRA to simulate these snow indicators at the

nine different locations, perform a time series analysis of the indicators and assess their spatial variability. The different locations have considerable spatial and temporal variability in snow cover characteristics and seasonal maximum snow water equivalent (amount of liquid water stored in the snowpack) values range from less than 50 mm w.e. to greater than 600 mm w.e. The correlation coefficients between maximum snow water equivalent output from Δsnow and CARRA/RACMO are 0.73 and 0.48 respectively. Correlation coefficients are highest for maximum snow water equivalent and snow cover duration,

and model and reanalysis output underestimate snow cover onset. We find little evidence of statistically significant ($p<0.05$) trends at varied periods between 1997 and 2021 except for the earlier onset of snow melt in Zackenberg (-8 days/decade, $p=0.02$, based upon RACMO output). While we stress the need for context-specific validation, this study suggests that in most cases snow depth or snow water equivalent output from CARRA can describe spatial-temporal characteristics of seasonal snow cover, particularly changes in melt onset and snow cover end.



## 1 Introduction

Seasonal snow cover has important climatic and ecological implications for the ice-free regions of coastal Greenland. Snow cover shows large spatial and temporal variability (Cohen, 1994; Rosen, 1992), thereby influencing local to regional climate variability by controlling the energy exchange between the surface and the atmosphere at different time scales from sub-seasonal to multi-decadal. The key mechanisms for this snow-atmosphere coupling are insulation of the ground (Cohen, 1994), the snow-hydrological effect (Preece et al., 2023), high surface albedo (Diro et al., 2018) and thermal emissivity (Warren, 1982). In addition, the snowpack needs significant amounts of energy for the melting process at the end of the snow season in spring (Henderson et al., 2018). Because of this role of snow cover in the climate system, it is an important factor in determining community and ecosystem structure in Arctic regions (Bokhorst et al., 2016; Bonsoms et al., 2024; Callaghan et al., 2012; Niittynen and Luoto, 2018; Walker et al., 1993). Interannual variability of snow cover characteristics as well as their long-term trends, thus, influence many relevant ecological processes (AMAP, 2011). It is well documented that seasonal snow cover is rapidly changing in the northern hemisphere (e.g. Brown and Robinson, 2011; Pulliainen et al., 2020) and especially throughout the Arctic. Examples of reported changes include a -4%/decade change in May snow cover extent (Derksen and Mudryk, 2023), significantly earlier occurring Northern Hemisphere snow melt (Foster et al., 2013), 3.4 days/decade earlier snow-free date in the Arctic (excluding Greenland) (Callaghan et al., 2012) and decreasing snow cover duration by 3-5 days/decade (Derksen et al., 2015). While snow cover onset trends in the Northern Hemisphere are generally negative (earlier), positive (later) trends exist as well, particularly in coastal regions (Allchin and Déry, 2020). Further changes in these variables are projected by climate models, for example, several studies estimate a 10-40% decrease in snow cover duration by 2050 (Bokhorst et al., 2016; Niittynen and Luoto, 2018). It is important to mention here that most remote sensing studies on changes in snow parameters exclude coastal Greenland, given the difficulty of accurately sensing these mountainous regions

Due to well-known challenges in snow monitoring (snow drift, high maintenance costs, lack of power supply, data gaps due to sensor failure), directly observed seasonal snow cover data from the ice-free coastal regions of Greenland is limited, owever, it does exist. Here, we present a little-used, quality-controlled dataset of seasonal snow, collected by Asiaq – Greenland Survey. We use this data to assess spatio-temporal characteristics of seasonal snow cover at each observation location in the ice-free regions of Coastal Greenland and assess whether state-of-the-art climate models can simulate these spatio-temporal characteristics. The presented data (section 2.1) originates from nine automatic weather stations distributed over the ice-free part of coastal Greenland (Figure 1). Six are located on the west coast and three are on the east coast. The in-situ data spans varying periods between 1997 and 2021 (see Table 1).



The aim of this study is to present, for the first time, a quality-controlled dataset of daily HS for the coastal regions of Greenland for the period 1997-2021. Using a simple modelling approach (Δsnow) we estimate snow water equivalent (SWE), which is the amount of liquid water stored in the snowpack, solely based on the daily time series of snow depth (HS). To overcome the shortcomings of the low temporal and spatial coverage of the daily HS observations, we then use the Δsnow output to evaluate the ability of a regional climate model (RACMOv2.3) and an Arctic reanalysis product (CARRA) to simulate observed spatio-temporal patterns of HS and SWE. Based on this validation, which is new for Greenland, we can then use the simulated HS/SWE to obtain spatio-temporal characteristics for several climatologically and ecologically relevant snow indicators (Callaghan et al., 2012; Lund et al, 2017; Wu et al., 2023): maximum SWE ($SWE_{max}$), start of melting ($M_{onset}$), duration of melting ($M_{duration}$), duration of snow cover ($SC_{duration}$), start of snow cover ($SC_{onset}$) and end of snow cover ($SC_{end}$) (Figure 2) for Greenland. In this way, ecologically and climatologically relevant differences in snow cover accumulation and depletion can be substantiated with data. In particular, we investigate whether there are systematic differences in snow cover characteristics between Greenland's east and west coasts and what factors explain these differences. We use different statistical measures to provide a first insight into the spatio-temporal differences and trends in seasonal snow cover, taking statistical significance into account.

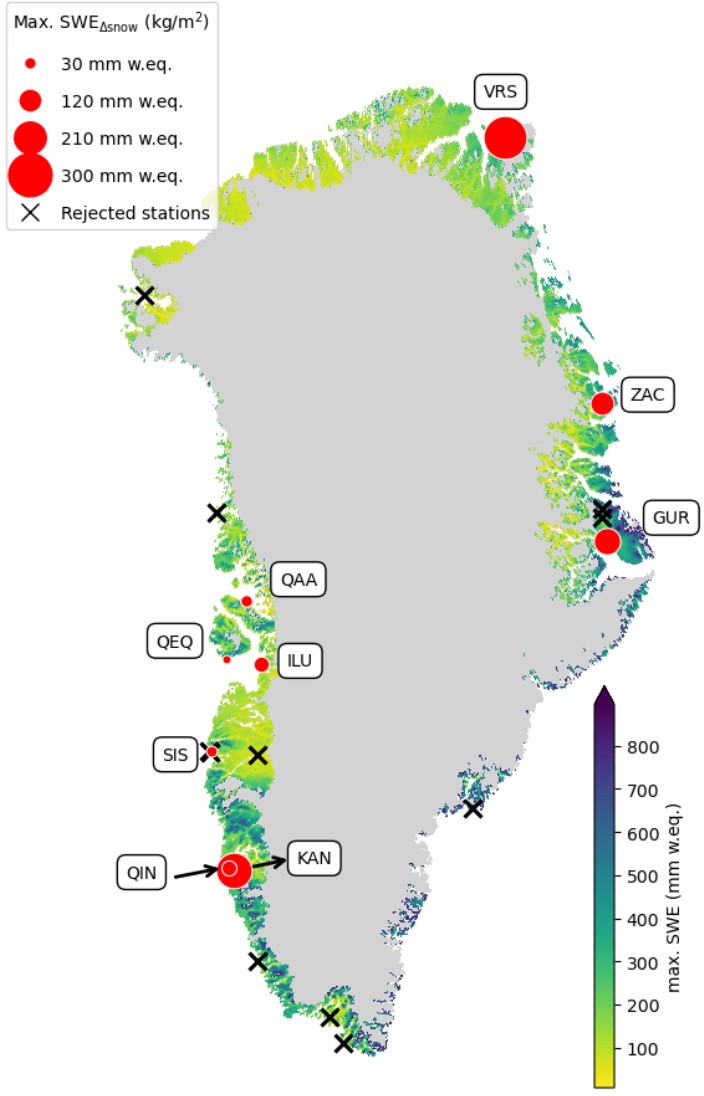

**Figure 1**: Asiaq weather station locations with snow observations in coastal The background map shows CARRA max. SWE averaged over the period 1990-2022. The size of the red dots indicates the average max. SWE$_{ΔSNOW.MODEL}$ (section 3.1) during the period with measurements.



## 2 Data

### 2.1 Snow depth observations (HS$_{OBS}$)

The snow depth (HS) data presented in this study were collected by Asiaq - Greenland Survey at their weather stations located along the west and east coasts of Greenland. From a total of 23 weather stations, nine are presented here. The remaining 14 are excluded due to data gaps, insufficient duration of the measuring period, or the fact that the resulting data did not accurately represent general local snow conditions (e.g., due to wind-related factors) (see rejected stations in Figure 1). The datasets span different periods within the period 1990 until 2021 (Table 1). HS has been measured using sonic ranging sensors (most commonly Campbell Scientific SR50a), which are known to have an accuracy of ca. +/- 1.0 cm (Campbell Scientific, 2021). These sensors are installed as part of the weather station which is set up to measure representative weather conditions in the towns and villages. Site selection for these weather stations was based on WMO recommendation, balanced with infrastructure and logistic requirements. The dataset of nine time series of HS used in this study will hereafter be referred to as HS$_{OBS}$.

To help interpretation of the measured values, we provide a basic climatology based on CARRA output for all locations that are part of the presented datasets (Table 1). The seasonal variables $T_{2m}$ (°C), modelled accumulated solid precipitation (mm w.e.) and wind speed (m/s) have been calculated from CARRA output for the period 1990-2022. Incomplete seasons at the beginning and end of the 1990-2022 period have not been considered in the calculation.

Two meter air temperature from reanalysis is considerably lower at the three weather stations on the east coast compared to the west coast. Specifically, the climatology of VRS (the most northerly station) stands out with respect to temperature when compared to the other locations. VRS shows the lowest temperature in each season and the annual temperature amplitude is largest at this location (29.3°C). Furthermore, VRS is the only location where the CARRA solid precipitation output in summer is still relatively high (81 mm w.e.). QIN and KAN, located at a much lower latitude, are the only other locations where the summer solid precipitation output is above 10 mm w.e. (11 and 33 respectively). Wind speed has a clear influence on the spatial variability of seasonal snow cover on a local scale. Greater small-scale spatial variability of HS is expected at locations with higher wind speeds (especially in winter when snow is easier to erode and drift). The CARRA average winter wind speeds at the selected nine locations range from 3.5 (at GUR) to 7.2 m/s (at ILU). Of all nine locations, the ZAC region is the most well-studied in terms of spatio-temporal characteristics of snow cover (e.g. Kankaanpää et al., 2018; Pedersen et al., 2016).





**Table 1:** CARRA climatology for each location

| Location and elevation | Observation stations | | CARRA climatology (1990 – 2022) | | |
|---|---|---|---|---|---|
| | Coordinates | Period with snow observations | $T_{2m}$ (ºC) DJF, MAM, JJA, SON | Annual solid/liquid precipitation (mm w.e.) DJF, MAM, JJA, SON | Mean. wind speed (m/s) DJF, MAM, JJA, SON |
| Villum Research Station (VRS) 37 m a.s.l. | 81.58°N 16.64°W | 2014/08/26 2018/08/08 | -26.6 -19.0 2.7 -15.2 | 98/0 83/0 81/11 140/9 | 3.7 3.3 3.3 3.6 |
| Zackenberg (ZAC) 44 m a.s.l. | 74.47°N, 20.55°W | 1997/07/01 2020/09/21 | -16.4 -10.9 5.5 -7.2 | 136/1 66/1 5/53 90/23 | 4.5 3.4 3.0 3.9 |
| Gurreholm (GUR) 80 m a.s.l. | 71.24°N 24.55°W | 2008/09/07 2010/08/19 | -16.5 -10.6 5.2 -6.5 | 120/0 71/4 3/69 89/33 | 3.5 2.6 2.0 3.0 |
| Qaarsut (QAA) 90 m a.s.l. | 70.74°N 52.71°W | 2008/07/01 2021/11/03 | 12.0 -8.4 6.2 -2.0 | 39/4 29/3 3/47 43/23 | 5.3 3.9 3.9 5.6 |
| Qeqertarsuaq (QEQ) 12 m a.s.l. | 69.24°N 53.53°W | 2018/07/01 2020/12/30 | -9.6 -6.6 6.9 0.1 | 96/9 75/5 5/117 97/60 | 5.6 4.0 3.7 5.6 |
| Ilulissat (ILU) 29 m a.s.l. | 69.24°N 51.06°W | 2008/07/01 2021/10/22 | -12.2 7.1 6.9 -2.4 | 58/6 51/10 4/82 64/39 | 7.2 4.9 3.6 7.0 |
| Sisimiut (SIS) 15 m a.s.l. | 66.94°N 53.69°W | 2008/07/01 2011/06/29 | -11.1 -6.0 6.4 -0.8 | 106/23 68/10 5/100 88/50 | 6.3 5.4 4.2 5.6 |
| Qinngorput (QIN) 38 m a.s.l. | 64.17°N 51.67°W | 2007/11/28 2009/11/19 and 2010/07/01 2011/09/27 | -7.8 -3.6 6.4 0.0 | 178/23 137/16 11/47 113/80 | 7.1 6.4 5.2 6.5 |
| Kangerluarsunnguaq (KAN) (also known as Kobbefjord) 40 m.asl | 64.13°N 51.34°W | 2008/07/01 2011/06/26 | -10.6 -5.8 5.9 -2.3 | 224/16 172/20 33/164 199/103 | 6.2, 5.5 4.6 5.9 |



## 2.2 Manual snow water equivalent observations (SWE_OBS)

The manual snow water equivalent observations are used for the evaluation of Δsnow (see 3.1). These measurements were collected as part of the Greenland Ecosystem Monitoring program (GEM) by Asiaq – Greenland Survey in KAN and Aarhus University in ZAC. Snow pit data have been collected from 2004. These snow pit measurements are known to generally have an accuracy of lower than 10%, however this number can reach 15% as well (López-Moreno et al., 2020). The dataset of snow pit measurements will hereafter be referred to as SWE_OBS.

## 2.3 CARRA reanalysis dataset (SWE_CARRA and HS_CARRA)


The Copernicus Arctic Regional Reanalysis (CARRA) (Schyberg et al., 2021) uses HARMONIE as a surface scheme including a snow model. The atmospheric assimilation uses a three-dimensional variational data approach. Surface variables are assimilated using an optimal interpolation approach. In this study, we use the CARRA-West domain, which includes Greenland. CARRA is laterally forced with ERA5 and produces a 3-hourly output on a 2.5-km horizontal grid space. Snow

output, as opposed to other variables like temperature, is not constrained by the assimilation of snow measurements from Asiaq and is purely a model product. Atmospheric variables observed at weather stations from Asiaq are assimilated. From the Greenland Ice Sheet, weather stations from GC-Net (Steffen et al., 1996; Steffen and Box, 2001) and PROMICE (Van As, 2011) are also part of the assimilation. Here we use two variables from the CARRA reanalysis datasets: the SWE variable, which will hereafter be referred to as SWE_CARRA and the snow density variable. The snow density variable is used in

combination with SWE_CARRA to calculate snow depth (which gives HS_CARRA, see methods section) for the period 1990-2023.

## 2.4 RACMO2.3p2

The Regional Atmospheric Climate Model (RACMO2) was developed by the Royal Netherlands Meteorological Institute (KNMI) (van Meijgaard et al., 2008). The polar version of this model, RACMO2.3p2, was developed to adequately simulate the evolution of surface mass balance over the ice sheets of Greenland, Antarctica and other glaciated regions. In this study,

we use the statistically downscaled product at 5.5 km (Noël et al., 2019). RACMO2.3p2 is forced on a three-hourly basis by ERA-40 (1958–1978), ERA-Interim (1979–1989) and ERA-5 (1990–2021). The topography in RACMO2.3p2 at 5.5 km spatial resolution is derived from the GIMP digital elevation model at 90 m downsampled to 5.5 km (Howat et al., 2014).

## 3 Methods

### 3.1 Δsnow

Δsnow (Winkler et al., 2021) is an algorithm that determines SWE, with the only input being daily values of HS. Despite this simplicity in terms of input, the model incorporates several complex snow climatological processes (e.g., compaction, melting and refreezing) from the daily changes in HS. Previous evaluation attempts for Δsnow in Arctic and mountain regions in the





Northern Hemisphere have shown promising results. The range of biases for Δsnow has been reported to be -15 to +17.2% ($SWE_{max}$) and -7.3 to 3.0 days ($M_{onset}$), using eight datasets from different regions (Fontrodona-Bach et al., 2023). Winkler et

al. have reported a bias of 0.3 mm w.e. and RMSE of 36.3 mm w.e. for $SWE_{max}$ (2021). The output from Δsnow will, hereafter, be referred to as $SWE_{\Delta snow}$. We evaluate the performance of Δsnow with $SWE_{OBS}$, giving the Pearson correlation coefficient (r), a measure of absolute error (RMSE) and the mean absolute percentage error (MAPE) as a measure of relative error.

**3.2 Evaluation of climate models used in this study (CARRA/RACMO)**

Both CARRA and RACMO2.3p2 output are evaluated using the $SWE_{\Delta snow}$ dataset. We evaluate several snow indicators that

we define in the following section (Figure 2). For these snow indicators, we determine the Pearson correlation coefficient (r), p-values (p) related to the statistical significance of the correlation coefficient and Root Mean Square Error (RMSE) for the correlation between $SWE_{CARRA}$/$SWE_{RACMO}$ and $SWE_{\Delta snow}$. The model values are obtained using the nearest grid cells based on the coordinates of the nine selected locations with snow observations. The model data was then resampled to match the temporal resolution and period from the observational data. The data was also restructured per hydrological year (1st of October

– 30th of September), whereafter snow indicators (Figure 2) for each hydrological year were calculated. Selected and pre-processed SWE values based on RACMO output will, hereafter, be referred to as $SWE_{RACMO}$. To directly compare CARRA output with $HS_{obs}$, we calculated HS from CARRA based on the SWE and snow density variables.

$$CARRA_{HS} = CARRA_{SWE} \div CARRA_{density} \quad (1)$$


$SWE_{RACMO}$ is estimated based on the variables available from the model:

$$SWE_{RACMO} = solid\ precipitation - snowmelt - sublimation - snow\ drift \quad (2)$$

This value is then corrected by adding the minimum value in the first hydrological year because the snow depth at the beginning of the first hydrological year is unknown. In the following years, the value of the last day of the previous hydrological is the starting snow depth.

**3.3 Analysis of spatio-temporal characteristics of seasonal snow cover**

Since our objective is to analyse whether state-of-the-art regional climate models can be used to assess changes in seasonal

snow cover that have relevance to the local ecosystems, we define six ecologically relevant snow indicators (Figure 2).




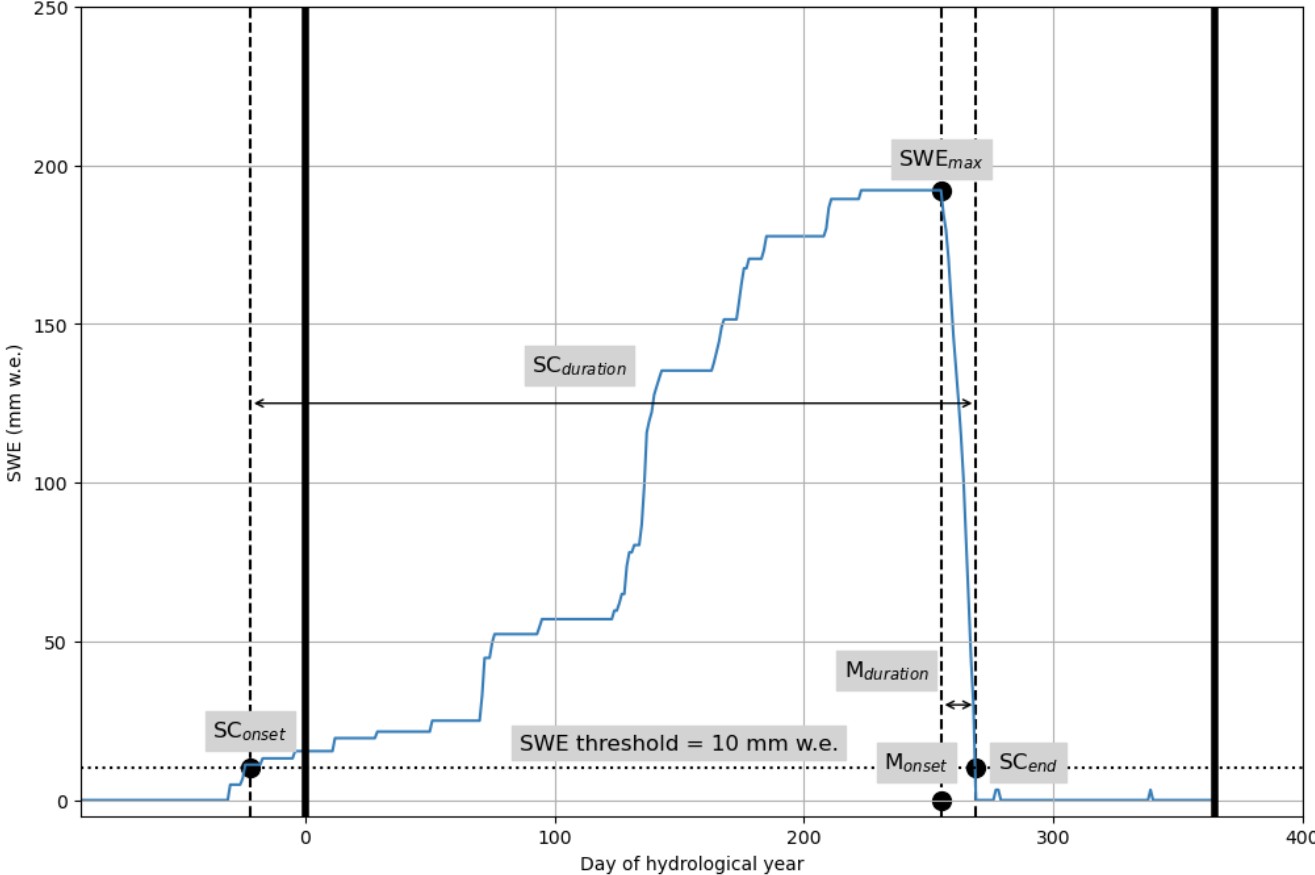

**Figure 2:** Definition of snow indicators. The SWE time series is from location ZAC 1998. The threshold of 10 mm w.e. is indicated with a dotted black line.

To ensure we capture the beginning of the snow buildup, which sometimes starts before October 1st (e.g., at ZAC, 1998), we include the last 90 days of the previous hydrological year when calculating snow indicators. Below we give a summary of the indicators used:

- **Max. SWE ($SWE_{max}$)** is the largest SWE of a particular hydrological year, calculated with a rolling mean of 5 days.
This rolling mean is used to reduce the impact of daily fluctuations and allows us to more accurately determine the peak SWE value within a season. The unit is 'mm w.e..'.

- **Melt onset ($M_{onset}$)** is the corresponding day of the hydrological year with $SWE_{max}$. In the case of multiple days with the same $SWE_{max}$ values, the last day with that value is chosen as $M_{onset}$. The unit is the 'day of hydrological year'.






- **Snow cover duration (SC$_{duration}$)** is here defined as the longest continuous period of SWE above the threshold of 10 mm w.e. within a specific hydrological year, including 90 days before that hydrological year. SC$_{duration}$ is normally defined using a snow depth threshold (e.g. Notarnicola, 2022). Here we use a SWE threshold because it is a variable that can be calculated from all three datasets. The unit for SC$_{duration}$ is 'number of days'.


- **Snow-cover end (SC$_{end}$)** is the last day of the longest continuous period of SWE above the threshold of 10 mm w.e. within a specific hydrological year, including 90 days before that hydrological year. The unit is the 'day of the hydrological year'.


- **Snow-cover onset (SC$_{onset}$)** is the first day of the longest continuous period of SWE above the threshold of 10 mm w.e. within a specific hydrological year, including 90 days before that hydrological year. The unit is the 'day of hydrological year'.

- **Melt phase duration (M$_{duration}$)** is the length of the period between the melt onset and the snow-cover end. The unit is 'number of days'.


### 3.4 Trend analysis

In addition to using the calculated snow indicators to evaluate the climate model output, we also assess trends in the annual values of these indicators for locations with five or more years of measurements with the Mann-Kendall trend test. Trends will be indicated with Theil-Sen estimators/slope and the intercept of the Kendall-Theil Robust Line. For further details see Hussain and Mahmud (2019).


### 4. Results

### 4.1 Evaluation of Δsnow

The output of the Δsnow model is evaluated with snowpit measurements at ZAC and KAN (Fig. 3). These measurements took place annually, with several measurements in each year, in ZAC starting from 2000 (cf. Fig. 4) and at KAN starting from 2013. They provide us with a detailed understanding of the performance of Δsnow, which derives SWE values at these two locations relatively well (r = 0.92 at ZAC and r = 0.82 at KAN). This comparison is most robust at ZAC, where it is based on more data points (n = 55) than at KAN (n = 9). The discrepancies between Δsnow and snowpit measurements (RMSE = 43.10 mm w.e. at ZAC and RMSE = 115.51 mm w.e. at KAN) are particularly evident for higher SWE values (above 300 mm w.e.) at both locations. Without more validation measurements, especially in these conditions with high SWE values, it is difficult to interpret the exceptionally large difference between Δsnow and the snowpit measurement at KAN in 2015 (where the measured SWE was 596 mm w.e.). Most probably it can be explained by the tendency of Δsnow to underestimate SWE for higher values.







However, considering the much higher relative bias compared to all other data points, measuring error can also not be excluded. The overall high correlation coefficients support our usage of Δsnow model values as a reference dataset for comparison with RACMO2.3p2 and CARRA.

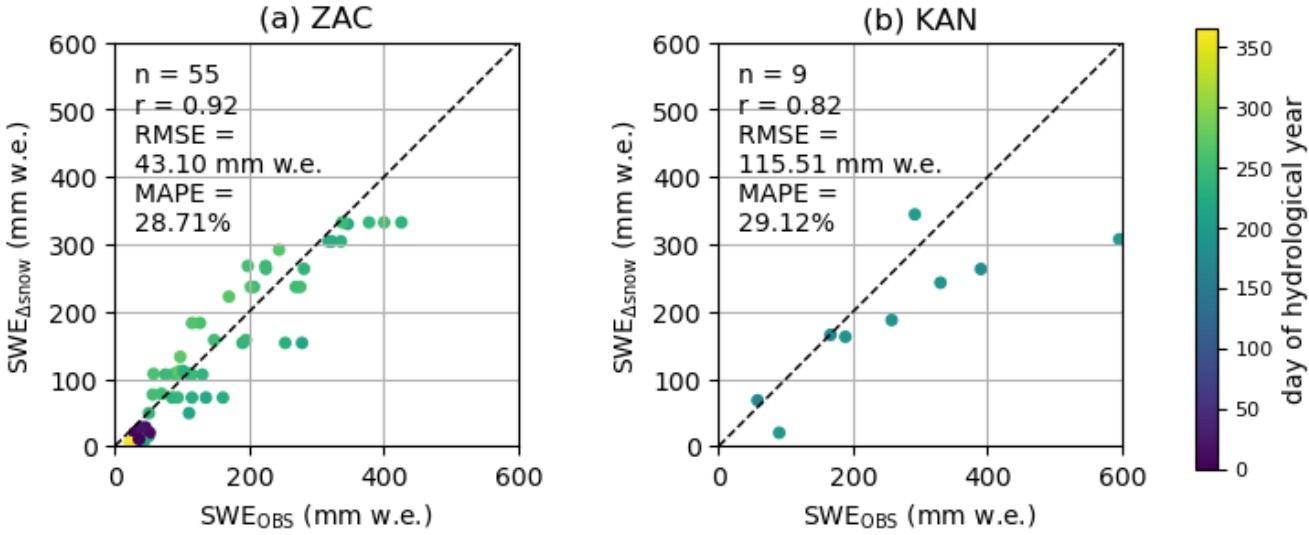


**Figure 3:** Evaluation of SWE$_{Δsnow}$ with measured SWE$_{OBS}$ at locations ZAC (a) and KAN (b). The day of the hydrological year for each data point is indicated by the colour of the dot.

**4.2 Evaluation of CARRA and RACMO2.3p2**

CARRA reanalysis and RACMO2.3p2 output are evaluated against Δsnow output using a daily correlation between these datasets (Figure 5) and the calculated snow indicators (see section 3.3) (Figure 6). The length of the period with HS$_{obs}$ (and therefore Δsnow output) available for this evaluation ranges from only two years (at QEQ and GUR) to 24 years (at ZAC). The correlation between SWE$_{Δsnow}$ and SWE$_{CARRA}$ is location-dependent, it varies between 0.44 and 0.90 (Figure 5). We find the highest Pearson correlation coefficient (r) at VRS, GUR and ZAC (r=0.87/0.90/0.90 respectively and p<0.01) and the

lowest Pearson correlation coefficient in QEQ (r=0.44, p<0.01). In general, the Pearson correlation coefficients between SWE$_{CARRA}$ and Δsnow are higher than the correlation coefficients between SWE$_{RACMO}$ and Δsnow (Figures 5 and 6). Correlation coefficients for SWE$_{max}$ are 0.73 and 0.48 for CARRA/RACMO respectively. SWE$_{RACMO}$ shows a similar or worse match with SWE$_{Δsnow}$ when compared to SWE$_{CARRA}$. When looking at all locations together, there is no clear over- or underestimation from either CARRA or RACMO2.3p2 when compared with Δsnow. Instead, at some locations, the models

show a clear overestimation of SWE values (at ZAC, KAN, QIN, SIS and QEQ), while at other locations, the SWE values are underestimated (at ILU and QAA). In general, the bias of RACMO and CARRA has the same sign (positive or negative) at each location. We notice a seasonal pattern in the data (e.g. at ZAC and KOB): SWE values are underestimated at the beginning of the year by both CARRA and RACMO2.3p2, while later on in the season they are overestimated (Figure 5).



The three locations with the highest latitudes (at VRS, ZAC and GUR) generally have more seasonal snow than those with

lower latitudes (Figure 4, Figure 5). While some locations show a negative correlation for some snow indicators, these values are never statistically significant at the 95% confidence level. For each snow indicator, there is a significant and positive correlation between $\Delta$snow and CARRA, except for $M_{onset}$ and $M_{duration}$ (Figure 6). The range of different correlation coefficients per location is relatively narrow for $SWE_{max}$ while it is much wider for the other indicators (Figure 7). RMSE values are highly dependent on the particular location. For example, a clear positive bias of SWE is present in both $SWE_{RACMO}$

and $SWE_{CARRA}$ for location ZAC (Figure 4) and this bias is stronger in $SWE_{RACMO}$. This particular location contributes the most to the overall RMSE of 114/188 mm w.e. for CARRA/RACMO2.3p2, respectively, due to the clear overestimation in combination with the relatively large contribution to the overall number of data points. CARRA output matches particularly relatively well with $SWE_{\Delta snow}$ values for the indicators $SWE_{max}$ and $SC_{end}$. $SC_{onset}$ is underestimated by the climate models and reanalysis output, and the spread is generally higher. There is a clustering of $SWE_{\Delta snow}$ $SC_{onset}$ values around 0, indicating that

snow cover often starts around the start of the hydrological year.










**Figure 4:** Time series of SWE$_{\Delta snow}$ (orange), SWE$_{CARRA}$ (blue) and SWE$_{RACMO}$ (green) for all nine locations in Greenland coastal ice-free regions. The time series of ZAC and KAN include manual SWE measurements (SWE$_{OBS}$) (red dots). Periods without SWE$_{\Delta snow}$ data are shaded light grey.

**Figure 5:** Daily SWE$_{\Delta snow}$ values against daily SWE$_{CARRA}$ values for each location. The colour of the dots indicates the day of the hydrological year, and the red line indicates the slope and intercept of the correlation. HS$_{\Delta snow}$ / HS$_{CARRA}$ and SWE$_{\Delta snow}$/ SWE$_{RACMO}$ plots can be found in Appendix A.



**Figure 6:** Evaluation CARRAs and RACMOs ability to simulate several snow indicators. Locations are identified with different colours and symbols. Open markers show values for RACMO calculated SWE output, while closed markers show values for CARRA reanalysis. Negative snow cover onset means snow cover onset before the start of the hydrological year (starting 1st of October).





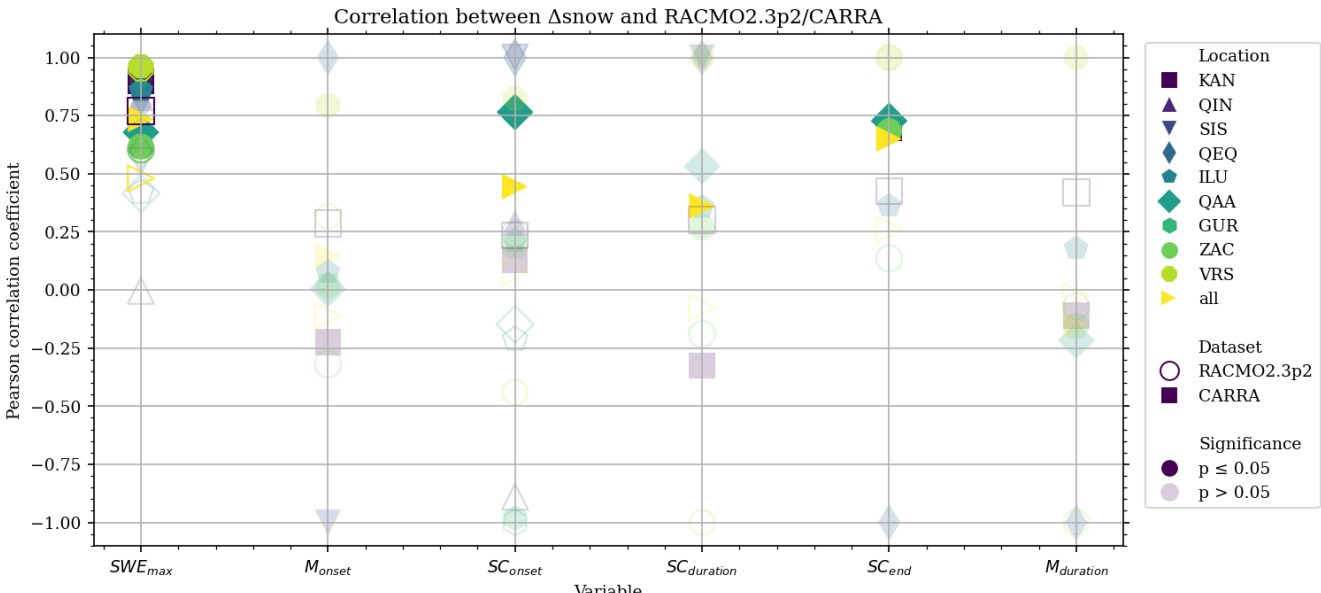

**Figure 7:** Pearson correlation coefficients (r) per snow indicator (x-axis), location (marker type) and dataset (open and closed markers). Non-significant correlations (P≤0.05) are transparent and significant correlations (p>0.05) are non-transparant.

### 4.3 Analysis of spatiotemporal characteristics of seasonal snow cover

There is considerable spatial and temporal variability between the different locations (Figure 4). Seasonal $SWE_{max}$ values range from less than 50 mm w.e. to greater than 600 mm w.e. In general, the interannual variability from $SWE_{\Delta snow}$ is also visible in the $SWE_{CARRA}$ and $SWE_{RACMO}$ datasets. However, the agreement between the datasets varies strongly from year to year and between the different locations. While most locations show realistic patterns related to the build-up and melt of seasonal snow cover, some exceptions are present in the dataset. The largest $SWE_{max}$ values are observed in VRS (averaged 354 mm w.e. from $SWE_{\Delta snow}$), located in the northeast of Greenland. Figure 4 also shows that the buildup of snow cover starts before the start of the hydrological year in the locations VRS and GUR. The average max. $SWE_{max}$ for all nine stations in Greenland, giving equal weight to each location, is 129 mm w.e. (highly variable, with a standard deviation of 106 mm w.e.), which gives a first number of average SWE for Greenland's coastal regions based on observations.

At the 95% confidence level, the only statistically significant trends in our datasets are for $M_{onset}$ at location ZAC for $SWE_{RACMO}$ (-8 days/decade, p=0.02) (Figure 8). In the $SWE_{OBS}$ and $SWE_{CARRA}$ datasets, these trends have a similar direction and magnitude (-8 days/decade, p=0.32 and -8 days/decade, p=0.18 respectively), but are not statistically significant at the 95% confidence level. $SWE_{max}$ is the snow indicator with the highest interannual variability (e.g. at ZAC: standard deviation = 99 mm w.e.) in the period 1998 – 2020). This interannual variability increases in the last years of the period (2013 – 2020). In this period, the consistency between the different datasets decreases, which means that the model and reanalysis products have difficulty accurately representing this increased interannual variability in these years.



**Table 2:** Descriptive statistics of $SWE_{max}$ (other indicators detailed in appendix B)

| Season | Statistic (mm w.e.) | KAN | QIN | SIS | QEQ | ILU | QAA | GUR | ZAC | VRS |
|---|---|---|---|---|---|---|---|---|---|---|
| DJF | Mean $SWE_{\Delta snow}$ | 88 | 23 | 35 | 15 | 43 | 37 | 165 | 64 | 223 |
| | Mean $SWE_{CARRA}$ | 166 | 108 | 73 | 72 | 45 | 22 | 119 | 164 | 219 |
| | Range/standard deviation $SWE_{\Delta snow}$ | 267/58 | 146/34 | 66/30 | 49/17 | 193/43 | 87/23 | 252/100 | 277/60 | 287/64 |
| | Range/standard deviation $SWE_{CARRA}$ | 384/78 | 138/32 | 157/52 | 88/22 | 110/25 | 46/10 | 128/38 | 539/93 | 291/69 |
| MAM | Mean $SWE_{\Delta snow}$ | 192 | 81 | 31 | 7 | 63 | 41 | 234 | 141 | 336 |
| | Mean $SWE_{CARRA}$ | 166 | 126 | 88 | 75 | 35 | 18 | 192 | 267 | 324 |
| | Range/standard deviation $SWE_{\Delta snow}$ | 349/111 | 239/94 | 79/36 | 34/10 | 247/63 | 108/28 | 256/102 | 333/90 | 210/71 |
| | Range/standard deviation $SWE_{CARRA}$ | 395/95 | 239/73 | 182/58 | 137/47 | 138/35 | 62/18 | 255/44 | 591/138 | 267/88 |
| JJA | Mean $SWE_{\Delta snow}$ | 8 | 3 | 0 | 0 | 1 | 3 | 33 | 29 | 165 |
| | Mean $SWE_{CARRA}$ | 8 | 3 | 0 | 0 | 1 | 3 | 33 | 29 | 165 |
| | Range/standard deviation $SWE_{\Delta snow}$ | 293/38 | 168/17 | 0/0 | 0/0 | 172/9 | 108/8 | 332/81 | 333/70 | 467/184 |
| | Range/standard deviation $SWE_{CARRA}$ | 243/29 | 2/0 | 1/0 | 11/1 | 11/0 | 3/0 | 271/80 | 637/107 | 488/143 |
| SON | Mean $SWE_{\Delta snow}$ | 7 | 1 | 7 | 2 | 6 | 10 | 36 | 7 | 62 |
| | Mean $SWE_{CARRA}$ | 19 | 15 | 12 | 9 | 7 | 5 | 23 | 32 | 65 |
| | Range/standard deviation $SWE_{\Delta snow}$ | 56/13 | 20/4 | 50/14 | 16/4 | 51/11 | 49/14 | 150/41 | 79/12 | 190/49 |
| | Range/standard deviation $SWE_{CARRA}$ | 171/30 | 99/27 | 93/20 | 40/13 | 77/12 | 27/6 | 78/24 | 180/38 | 203/52 |





**Figure 8:** Time series of snow indicators for the locations ZAC and KAN. Trend statistics from the Mann-Kendall non-parametric test, Theil-Sen's Slope Estimator and the intercept of the Kendall-Theil Robust Line are displayed in the text box for each dataset. The year is abbreviated as 'yr' and day of hydrological year is abbreviated as 'dohy'.



## 5. Discussion

### 5.1 Model performance

One of the objectives of this study was to assess the performance of state-of-the-art climate models in simulating ecologically and climatologically relevant spatio-temporal characteristics of snow indicators. Our results have shown that CARRA reanalysis output is generally better suited for this purpose than RACMO2.3p2. As shown in Figure 6, the performance of both products is dependent on the specific snow indicator and study site. Our results suggest that CARRA reanalysis can especially be a useful tool for studying spatio-temporal trends in $SWE_{max}$ and $SC_{end}$, which have the highest correlation coefficients in our comparison with the $SWE_{\Delta snow}$ values (0.73 and 0.65 respectively).

The comparison of $HS_{CARRA}$, or $SWE_{CARRA}$ with snow observations as in our study is a rather novel approach and a limited amount of other studies can be found with similar methodologies. Maniktala (2022) has compared CARRA reanalysis output with snow observations for three low-precipitation sites in Svalbard, and found correlation coefficients of 0.74/0.58/0.46, RMSE values (m) of 0.07/0.14/0.08 and biases (m) of -0.06/-0.15/-0.02 in Hornsund, Ny-Alesund and Svalbard Airport, respectively. The average of the nine correlation coefficients for the locations in coastal Greenland is 0.76 (with a min. of 0.44 in QEQ and a max. of 0.90 in ZAC and GUR) (Figure 5) and is thus higher than for the Svalbard locations. Similar to what was reported for Svalbard, we notice generally higher correlation coefficients for areas with higher amounts of winter precipitation, which are mostly located on the east coast of Greenland. We suggest this is likely the main reason for the lower correlation between $SWE_{\Delta snow}$ and RACMO2.3p2/CARRA in some locations (e.g. at QEQ and QIN) as these mentioned locations have low average $SWE_{max}$ values (38 and 20 mm w.e., respectively). The snow cover in areas with higher average $SWE_{max}$ values is less sensitive to variability in solid precipitation and wind because the changes resulting from variability in these parameters are relatively small in a dense snowpack. The higher correlation values in high-precipitation locations show potential for using CARRA reanalysis output in water balance studies.

While not directly comparing reanalysis output to measured snow data, Krampe et al. (2023) used ERA5 reanalysis to force the snow model Crocus (Vionnet et al., 2012) for location VRS and concluded that Crocus has the potential to adequately represent snow depth evolution at this site. We have shown here that both RACMOv2.3 and CARRA reanalysis can simulate snow conditions in this region relatively well in the period 2014-2018 and seem to perform better than Crocus simulations forced with ERA5, as shown in (Krampe et al., 2023). While we stress the need for context-specific validation, we suggest that using $HS_{CARRA}$ or $SWE_{CARRA}$ directly can in some situations be a suitable alternative for snow model simulations forced with reanalysis data.

### 5.2 Drivers of spatial variability in seasonal snow cover

The seasonal snow cover datasets presented in this study are characterised by significant spatial and interannual variability of snow indicators, which have been quantitatively reported with descriptive statistics (see Table 2). While the number of locations with snow observations in ice-free Greenland is limited, the locations used in this study cover a wide range of





geographical and climatological conditions (see Table 1). These differences likely contribute to the spatial variability of the snow indicators, given the known influence of winter solid precipitation (Buus-Hinkler et al., 2006; Farinotti et al., 2010; Ide and Oguma, 2013; Kepski et al., 2017; Pedersen et al., 2018), winter temperature (McCabe and Wolock, 2010), wind variables and radiation and heat fluxes on snow conditions (Mott et al., 2018). Temperature trends in the ice-free regions of Greenland are characterised by strong seasonal and regional variability (Hanna et al., 2012; Zhang et al., 2022). The differences in local climate are, apart from local influences like topography, largely governed by large-scale circulation patterns. For example, large-scale atmospheric circulation patterns most commonly advect moisture from the North Atlantic Ocean to the East coast of Greenland. In particular, East Greenland is located near the North Atlantic storm track. This is an important factor explaining higher rates of snowfall and, thus, enhanced seasonal snow cover along the east coast compared to the west coast (Hinkler et al., 2008). This effect can also be seen in our $SWE_{OBS}$ dataset, where each of the east coast locations has higher average $SWE_{max}$ values than the west coast locations (except for KAN).

KAN is characterised by relatively high $SWE_{max}$ values compared to other west coast locations. This highlights the large interregional variability in Greenlandic fjord systems. For example, QIN (located in proximity to KAN) is characterised by much smaller $SWE_{max}$ values. However, this is based on a short period of snow measurements at QIN, and a low correlation between the three datasets at this location. It should be noted that the QIN data should not be overinterpreted, as the weather station is not located ideally and the time series is relatively short. The QIN location is more exposed to southerly weather and, additionally, it is located below a steep rock wall. We still decided to include this location in our selection of weather stations as it highlights smaller-scale spatial variability.

One advantage that any model (and in particular the CARRA reanalysis product) has over point-based measurements is the smoothed representation of reality. Given the fact that we know that snow depth and SWE have high spatial variability on small spatial scales, the 2.5 km by 2.5 km resolution from CARRA is for some applications more useful than a point-based measurement. For example, there might be considerable variation in $SC_{end}$ within a 2.5 by 2.5 km area, due to topographical differences. The relatively high correlation between $SC_{end}$ based on $SWE_\Delta$snow and $SWE_{CARRA}$ (0.65, p<0.01, Figure 6) might indicate that the measured $SC_{end}$ is an accurate representation of the $SC_{end}$ in the immediate surroundings of the weather station, which is in line with what is reported by the GEMP (Christensen and Arndal, 2023). Even though the interannual variability of $M_{onset}$ at ZAC is high (standard deviation = 35 days in $SWE_{OBS}$), spatial patterns are consistent from year to year (Pedersen et al., 2016). Similarly, in KAN, the wind directions and the topography cause highly heterogeneous patterns of snow accumulation every year (Myreng et al., 2020). In general, variability in local-scale snow conditions is mainly driven by topographical variability (Dobrowski, 2011).

While increased poleward moisture transport from lower latitudes certainly plays a role, reductions in sea ice that allow greater evaporation from the ocean surface have been identified as the key driver for increased Arctic precipitation (Bintanja and Selten, 2014; Kopec et al., 2016). Due to its proximity to areas characterised by relatively high fractional sea ice cover, it is expected that decreasing sea ice will enhance snow accumulation in northeast Greenland (Bintanja and Selten, 2014). Considering the locations of the weather stations used in this study, it is likely that the east coast stations are more susceptible



to sea ice variability and trends. A smaller sea ice fraction over the Greenland Sea allows cyclones to move northwards, which causes more precipitation at higher latitudes (Sellevold et al., 2022).

On a local scale, sea ice has recently been indicated as an important driver of the climate at ZAC (Shahi et al., 2023). Specifically, a low sea ice fraction in the Greenland Sea has been directly linked with more solid precipitation in the region in
all seasons except summer. This confirms that sea ice variability influences seasonal snow cover variability at ZAC. We further hypothesize that sea ice variability in Hudson Bay could influence seasonal snow cover in northwest Greenland as it is known to influence the coastal climate in that region (Ballinger et al., 2020). Similarly, variability in the occurrence and size of polynyas in the Arctic Ocean could explain part of the relatively high seasonal snow depth values we observe at location VRS. Especially when compared to seasonal snow cover at other places in North Greenland (cf. Pedersen et al., 2016), this east coast
location has a relatively high HS/SWE output. While it is known that enhanced evaporation over polynyas can lead to increased snowfall over adjacent coastlines (Schneider and Budeus 1997, Maqueda et al. 2004), the exact strength of the mechanism has never been quantified. Here, we hypothesize that spatio-temporal characteristics of the North East water polyna could be an important factor that influences the enhanced seasonal snow cover at VRS.

One of the key spatial characteristics we reported is enhanced seasonal snow cover at higher latitudes in coastal Greenland,
which implies more snowfall at higher latitudes. This can mostly be explained by the fact that the east coast locations in our dataset have a higher latitude than the west coast locations. Therefore, the reported latitudinal trend in seasonal snow cover could also be related to the fact that the east coast receives more snowfall than the west coast.

### 5.3 Temporal variability, trends and ecosystem implications

Generally, in the Northern Hemisphere, snow cover duration and snow cover extent have been decreasing over the last 40 years (Box et al., 2019; Brown and Robinson, 2011; Meredith and Sommerkorn, 2019), but changes on a local scale do not indicate clear trends (Buchelt et al., 2022). In the future, climate models generally predict a further decrease in $SWE_{max}$ and $SC_{duration}$ for southern Greenland. For northern Greenland, models predict a slightly higher $SWE_{max}$, later $M_{onset}$ and a longer $M_{duration}$ (Hinkler et al., 2008). In global mountain regions, a negative trend of -3.6% snow cover extent and -15.1 days +/- 11.6
405 days $SC_{duration}$ has been reported over 38 years based on a combination of satellite data and model output (Notarnicola, 2022). This study, similar to others (e.g. Pedersen et al., 2016) does also not indicate clear past trends in any of the snow indicators (Figure 8). The only exception to this statement is earlier snowmelt occurring at ZAC, however, this trend is only significant for the $SWE_{RACMO}$ dataset (-8 days/decade, p = 0.02). Earlier snowmelt at ZAC could significantly alter tundra vegetation, as snow melt is an important driver of tundra spring phenology (Assmann et al. 2019). Earlier snowmelt occurring at ZAC has
410 been reported by Kankaanpää et al. (2018), who also reported earlier snowmelt in the period 1998-2014. They also stated a less clear pattern during the second part of their study period (2006-2014), which fits with our results. In the same study, it was reported that locations with an earlier snow cover end date show stronger trends in melt onset, whereas locations with a later snow cover end date show more variable trends in melt onset (Kankaanpää et al., 2018). The variability of seasonal snow cover is generally adequately explained by changes in the climate system (Thackeray et al., 2019). Long-term changes in snow




cover characteristics can significantly influence permafrost (Callaghan et al., 2012). Snow-permafrost interaction is important
for several of the locations in our study, as many are characterised by permafrost occurrence. SIS and ILU have discontinuous
permafrost, whereas the others (except QIN and KAN) have continuous permafrost (Christiansen and Humlum 2000). The
snow indicators $SC_{onset}$, $SC_{end}$ and $SC_{duration}$ are all particularly relevant for permafrost-related processes as they influence the
ground thermal regime (Callaghan et al., 2012). For example, a decreasing $SC_{duration}$ could increase permafrost thaw as in snow-
free conditions, the sensible heat flux becomes an energy sink during most of the season (Lund et al., 2017). Shorter $SC_{duration}$
is furthermore likely to increase plant productivity and carbon capture in areas with enough soil moisture (Callaghan et al.,
2012).

## 5.4 Limitations

While CARRA reanalysis has SWE values available as a model output, this output is not available from RACMO2.3p2. Future
versions of RACMO will directly have output available from an integrated snow model. We have chosen to include
RACMO2.3p2 in this comparing exercise to quantify the added value of a high-resolution reanalysis product such as CARRA,
as opposed to using atmospheric models without an integrated snow scheme.

## 6. Conclusions

We have presented a first insight into spatio-temporal characteristics of seasonal snow cover in Greenland's coastal regions
for the period 1997-2021. The conversion of the newly presented quality-controlled $HS_{obs}$ dataset to $SWE_{\Delta snow}$ has shown
promising results when tested against $SWE_{obs}$ from ZAC and KAN (r = 0.92 and r = 0.82 respectively).
We have shown that the high-resolution reanalysis dataset CARRA performed better than the atmospheric climate model
RACMOv2.3 when it comes to simulating spatio-temporal characteristics of ecologically and climatologically relevant snow
indicators in the ice-free regions of coastal Greenland. CARRA is particularly successful in simulating the snow indicators
$SWE_{max}$ and $SC_{end}$ (correlation coefficients are 0.73 and 0.65 respectively).
Our results underscore the potential of directly using HS and SWE output from reanalysis products for detailed snow studies,
especially in regions with high precipitation amounts where the correlation between $HS_{CARRA}$ and $HS_{\Delta snow}$ was highest. High-
resolution reanalysis products like CARRA have the additional benefit that they can capture local-scale snow variability. In
contrast, point-based measurements do not provide information about this local-scale variability which can be significant in
the complex topographical settings of the coastal regions of Greenland. While the observations presented in this study are a
much-needed addition to the available snow cover data in the remote Arctic region, we also emphasise the need for extending
upon current monitoring activities, particularly given the broad scale of climatic and geographical conditions present along
Greenland's coasts.
Significant spatial and interannual variability has been found in all three datasets. These findings highlight the wide range of
geographical and climatological conditions present along the ice-free regions of the Greenlandic coast. The spatial variability

in snow conditions also exists on a smaller spatial scale, as highlighted by the differences in values of snow indicators between QIN and KAN. In $SWE_{\Delta SNOW,}$ as well as in $SWE_{CARRA}$ and $SWE_{RACMO}$, higher $SWE_{max}$ values were found on the east coast of Greenland. Given our chosen station locations, this coincides with the fact that higher $SWE_{max}$ values are found at higher latitudes because the locations on the east coast have higher latitudes compared to those on the west coast. We hypothesised

that the higher $SWE_{max}$ values on the east coast of Greenland compared to the west coast can be attributed to its proximity to the North Atlantic storm track and large-scale atmospheric circulation patterns.

Even though large-scale studies have indicated changing snow conditions in the Arctic, we did not find many significant trends in our time series analysis of the snow indicators for Greenland's coastal regions. The only statistically significant trend (at the 95% confidence level), was an earlier occurrence of melt onset at ZAC based on the $SWE_{RACMO}$ dataset. Earlier melt onset

at ZAC is in agreement with previous studies in that region.

**Code availability**

The Python code used for producing the figures is available upon request from the first author.

**Data availability**

Output from CARRA can be downloaded from the Copernicus Climate Data Store ([https://climate.copernicus.eu/climate-data-store](https://climate.copernicus.eu/climate-data-store)). Output from RACMO2.3p2 is available upon request from the Institute of Marine and Atmospheric Research (IMAU). Currently, the snow data from Asiaq – Greenland Survey cannot be made publicly available. The data have been shared with the reviewers of this paper. An update will be provided if the data will be publicly available in the future.

**Short summary**

We present snow data from nine locations in coastal Greenland. We show that a reanalysis product (CARRA) simulates seasonal snow characteristics better than a regional climate model (RACMO). CARRA output matches particularly well with our reference dataset when we look at the maximum snow water equivalent and the snow cover end date. We show that seasonal

snow in coastal Greenland has large spatial and temporal variability and find little evidence of trends in snow cover characteristics.

**Author contribution**

JS, JA and WS were involved in the study design, which was carried out by JS. JS performed the data analysis, sometimes

supported by TS. JS prepared the manuscript with contributions from all co-authors.

**Competing interests**

The authors declare that they have no conflict of interest.



## Acknowledgements

This article has been written as part of the research projects Snow2Rain (funded by the Austrian Academy of Sciences) and Snow2School (funded by the Federal Ministry of Education, Science and Research). AI tools have sporadically been used as an assisting tool during the data analysis conducted for this paper.

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



**Appendix A**



**Figure A1:** Daily $HS_{\Delta snow}$ values against daily $HS_{CARRA}$ values for each location. The colour of the dots indicates the day of the hydrological year, and the red line indicates the slope and intercept of the correlation.



**Figure A2:** Daily SWE$_{\Delta snow}$ values are plotted against SWE$_{RACMO}$ values for each location. The colour of the dots indicates the day of the hydrological year, and the red line indicates the slope and intercept of the correlation.




## Appendix B

Table B1: Descriptive statistics of the snow indicator $M_{onset}$

| Season | Statistic | VRS | ZAC | GUR | QAA | QEQ | ILU | SIS | QIN | KAN |
|---|---|---|---|---|---|---|---|---|---|---|
| Year | Mean SWE$_{\Delta snow}$ (mm w.e.) | 256 | 218 | 217 | 177 | 107 | 193 | 279 | 41 | 208 |
| | Mean SWE$_{CARRA}$ (mm w.e.) | 236 | 214 | 217 | 154 | 182 | 165 | 171 | 224 | 172 |
| | Range/standard deviation SWE$_{\Delta snow}$ (mm w.e.) | 21/11 | 308/60 | 0/- | 188/61 | 12/8 | 195/49 | 124/88 | 0/- | 69/20 |
| | Range/standard deviation SWE$_{CARRA}$ (mm w.e.) | 15/8 | 88/26 | 0/- | 119/41 | 11/8 | 110/35 | 22/16 | 0/- | 69/22 |

Table B2: Descriptive statistics of the snow indicator $SC_{duration}$

| Season | Statistic | VRS | ZAC | GUR | QAA | QEQ | ILU | SIS | QIN | KAN |
|---|---|---|---|---|---|---|---|---|---|---|
| Year | Mean SWE$_{\Delta snow}$ (no.of days) | 290 | 189 | - | 176 | 58 | 131 | 92 | 21 | 188 |
| | Mean SWE$_{CARRA}$ (no. of days) | 288 | 249 | - | 138 | 184 | 153 | 190 | 208 | 207 |
| | Range/standard deviation SWE$_{\Delta snow}$ (no. of days) | 65/46 | 294/80 | -/- | 177/59 | 81/57 | 154/56 | 184/130 | 0/- | 114/35 |
| | Range/standard deviation SWE$_{CARRA}$ (no. of days) | 35/25 | 73/19 | -/- | 152/38 | 11/8 | 80/25 | 17/12 | 0/0 | 91/28 |

Table B3: Descriptive statistics of the snow indicator $SC_{onset}$

| Season | Statistic | VRS | ZAC | GUR | QAA | QEQ | ILU | SIS | QIN | KAN |
|---|---|---|---|---|---|---|---|---|---|---|
| Year | Mean SWE$_{\Delta snow}$ (day of hydrological year) | -12 | 52 | - | 44 | 64 | 79 | 44 | 206 | 49 |
| | Mean SWE$_{CARRA}$ (day of hydrological year) | -25 | 10 | -/ | 53 | 35 | 50 | 32 | 31 | 24 |
| | Range/standard deviation SWE$_{\Delta snow}$ (no. of days) | 48/34 | 164/46 | -/- | 168/45 | 62/44 | 190/52 | 0/- | 0/- | 153/44 |
| | Range/standard deviation SWE$_{CARRA}$ (no. of days) | 22/16 | 57/16 | -/- | 80/26 | 20/14 | 73/20 | 1/1 | 0/- | 42/16 |


Table B4: Descriptive statistics of the snow indicator $SC_{end}$

| Season | Statistic | VRS | ZAC | GUR | QAA | QEQ | ILU | SIS | QIN | KAN |
|---|---|---|---|---|---|---|---|---|---|---|
| Year | Mean SWE$_{\Delta snow}$ (day of hydrological year) | 278 | 260 | - | 220 | 122 | 211 | 228 | 227 | 238 |
| | Mean SWE$_{CARRA}$ (day of hydrological year) | 262 | 259 | - | 191 | 222 | 203 | 222 | 239 | 231 |
| | Range/standard deviation SWE$_{\Delta snow}$ (no. of days) | 17/12 | 39/11 | -/- | 218/51 | 19/13 | 169/44 | 0/- | 0/- | 60/17 |
| | Range/standard deviation SWE$_{CARRA}$ (no. of days) | 13/9 | 36/9 | -/- | 150/42 | 9/6 | 65/22 | 18/13 | 0/- | 64/21 |



Table B5: Melt duration

| Season | Statistic | VRS | ZAC | GUR | QAA | QEQ | ILU | SIS | QIN | KAN |
|--------|-----------|-----|-----|-----|-----|-----|-----|-----|-----|-----|
| Year | Mean $SWE_{\Delta snow}$ (no. of days) | 24 | 41 | - | 44 | 14 | 31 | 11 | 186 | 30 |
| | Mean $SWE_{CARRA}$ (no. of days) | 25 | 46 | - | 44 | 37 | 38 | 51 | 15 | 65 |
| | Range/standard deviation $SWE_{\Delta snow}$ (no. of days) | 4/3 | 228/47 | - | 180/53 | 7/5 | 133/36 | 0/- | 0/- | 58/18 |
| | Range/standard deviation $SWE_{CARRA}$ (no. of days) | 2/1 | 87/23 | -/- | 118/36 | 20/14 | 90/28 | 4/3 | 0/- | 81/31 |