# Peer review of "Seasonal snow cover indicators in coastal Greenland from in-situ observations, a climate model and reanalysis"

_EGUsphere, 2024_

## Author Comment (AC1)

**Response to reviewer 1**

We thank the anonymous reviewer for their valuable comments regarding the manuscript. Below, we outline how we intend to improve the manuscript based on each specific point that was brought forward. Reviewer comments are shown in red with an author response underneath.

Line 53, a spelling error, it should be "however".

- Line 53: We change the misspelt word 'owever' to 'however'.

Line 57-59, it seems better to put this content in the section of data.

- Lines 57-59: We accept the suggestion to remove these lines from the introduction section, where they were indeed somewhat misplaced.
  The information from L57-59 should be included in section 2.1 Specifically, we suggest the following changes:
  L94: Insert 'automatic' to specify the type of weather stations
  L98: Change '1990' to '1997'. The period mentioned in L57-59 was correct, but the period mentioned in section 2.1 was incorrect.
  L98: Insert the following sentence: "All stations are located in the ice-free part of coastal Greenland. Six are located on the west coast and three are located on the east coast.". This information was mentioned in L57-59. However, it was missing in the data section.

Line 60, what is the meaning of HS, which appears here for the first time.

- L60: The abbreviation HS was erroneously introduced in L65, which is the second occurrence of HS and not the first one. We suggest changing 'snow depth' to 'HS' in L65 and introducing the abbreviation HS in L60.

Line 136, can you give readers a bit more information regarding the snow simulation module in the RACMO2.3p2 dataset? Which physical processes are considered?

- Line 136: As mentioned in the manuscript (e.g. L424), the atmospheric model RACMO2.3p2 does not have snow output available and snow processes on the ground are not considered. To make this more clear, we suggest the following changes in section 2.4:
  L144: Add: 'As this atmospheric model does not include a snow simulation, we estimate SWE values based on atmospheric output, as detailed in section 3.2.'

Line 158, what is the method of resampling?

- Line 158: We suggest changing the sentence on L158-L159 to: "The CARRA data was resampled from 3-hour to daily time steps using mean values to match the observational data's temporal resolution." In this way, the exact method and model product are specified.

Line 185, why is the window of rolling means 5 days? How about 3 days or 7 days?

- Line 185: The window of the rolling mean should be as short as possible (to still capture the changes in snow evolution throughout the season), whilst removing inaccurate representations of the snow cover end date. The length of 5 days is ideal for the window of the rolling mean as 3 days would not remove all short-scale variability that influences the determination of the snow cover end date, and 7 days would introduce unnecessary smoothing. The number of 5 days was based on visual inspection of the snow cover evaluation in each season and location. We suggest mentioning this by adding the following sentence at L186: "We choose a window length of 5 days to remove misrepresentation of the snow cover end date due to daily variability, whilst at the same time not introducing unnecessary smoothing of the seasonal snow evolution (which would be the case with a longer window length)."

**Response to reviewer 2**

We thank the anonymous reviewer for their valuable comments regarding the manuscript. Below, we outline how we intend to improve the manuscript based on each specific point that was brought forward. Reviewer comments are shown in red with an author response underneath.

The paper would be easier to follow if the same symbols/colours were used to represent each site/dataset across the different figures. To give a few examples, the colour scale goes from blue as the start of the year and yellow as the end of the year in Figure 3, but the reverse is the case for Figure 5; The line styles for the 3 different datasets change between Figure 4 and 8; and different colours and symbols are used to represent the sites in Figures 6 and 7.

The units and notation used to refer to snow water equivalent is not consistent throughout the paper. This is usually referred to in mm w.e., but sometimes w. eq.. I think it would be more in line with other publications to just use mm for SWE. Also on notation, the term "snow cover end" feels somewhat unconventional and does not clarify that that is a date. Consider substituting for something like "snow cover end date" or "snow off date".

We understand these comments regarding readability and consistency throughout the paper and thank the reviewer for their suggestions. We want to incorporate the following improvements, many of which the reviewer also suggested:

- The colour bar of Figure 5 will be updated to match the style used in Figure 3. Contrary to what was mentioned by the reviewer, blue colours are used in both figures to indicate the start of the hydrological in the current version of the manuscript. However, as Figure 3 used integer numbering, while Figure 5 used a text label for each month, the conciseness can be improved. Hence, our solution is to use integer numbers as labels in the colour bar in Figure 5.

- As the reviewer correctly pointed out, the line styles indicating the different datasets in Figures 4 and 8 were inconsistent, and we have prepared revisions for the figures so that the line styles belonging to the respective datasets are the same in both figures.
- Similarly, we have prepared revisions for Figures 6 and 7 so that the colours and symbols used to represent the different sites are consistent.

- Regarding the unit for SWE, we suggest using 'kg/m$^2$'. The units 'mm', 'mm w.e.', 'mm w.eq.' and 'kg/m$^2$' are all used throughout the scientific literature. We turn to the WMO guidelines (WMO, 2018) which state that 'kg/m$^2$' should be used for snow water equivalent.

- We accept the reviewer's suggestion to change "snow cover end" to "snow cover end date" to make it clearer that the term indicates a specific date.

Regarding the minor/technical comments, please find our response below:

Line 45: Remove "occurring"

- 45: 'Occuring' removed

Line 53: Typo – however?

- L53: We change the misspelled word 'owever' to 'however'.

Line 101: Recommendations, not recommendation.

- L101: We change 'recommendation' to 'recommendations'.

Line 106: Please rephrase to clarify if the fractional seasons from 1990 and 2022 have been included in the calculation.

- L110: We changed the sentence to: "The fractional seasons (winter 1990 and 2020) at the beginning and end of the 1990-2022 period have not been considered in the calculation."

Table 1 (approx. line 117): Why is one of the stations underlined?

- Table 1 (approx. line 117): The underlining under 'Kangerluarsunnguaq' has been removed.

Line 123: Is it more likely to be an accuracy of 10% or 15%?

- L123: The authors of (López-Moreno et al., 2020) state: "Uncertainty induced by instrumental bias was generally less than 10% but it can reach 15%." In the manuscript, we mistakenly wrote that the measurements generally have an *accuracy* of lower than 10%, while instead, they have an *uncertainty* of less than 10%. I shall remove the half-sentence where I mention that they can reach 15% as well, as this is not vital and can understandably lead to confusion.

  L126: These snow pit measurements are known to generally have an uncertainty of lower than 10% (López-Moreno et al., 2020).

Line 126: What is HARMONIE? Please write out acronyms in full at first use

- Line 126: HARMONIE is the model name. While this name originates from the consortium 'Hirlam Aladin Research towards Mesoscale Operational NWP in Europe', the model is never indicated in this way throughout the literature. 'HARMONIE' should thus not be seen as an abbreviation, but rather as the full name of the model that originates from the half sentence given above. As the model is usually indicated as 'HARMONIE' in the scientific literature (e.g. Mottram et al., 2021; Yu et al., 2020), we would prefer to keep this sentence as originally written.

Line 140: Insert "resolution" after 5.5 km

- Line 140: Inserted "resolution" after 5.5 km

Lines 144 – 150: Please expand the description of the △snow model

- Lines 144 – 150: We expanded the description of delta.snow model, which now includes a brief overview of the core idea of delta.snow and a simple summary of the key modelling approach.

  Text that we suggest to add: "Δsnow (Winkler et al., 2021) is an algorithm designed to estimate SWE using solely daily values of HS as input. Despite its simple input requirements, the basic layer model incorporates complex snow processes such as compaction, melting and drenching, based on the daily changes in HS. This approach enables the derivation of the hydrologically and climatologically more relevant SWE variable from the more widely available HS records. In our case, it facilitates the comparison of the climate model output from RACMO2.3p2 and HS observations at weather stations. Compaction is treated following the principles of Newtonian viscosity and melting snow is distributed stepwise from top to bottom throughout the simulated layers of the snowpack. A detailed description of the physics of Δsnow can be found in Winkler et al. (2021)."

Line 181: Add reference to Fig 2

- Line 181: The reference to Figure 2 is given in line 180, one sentence earlier. Hence, we would prefer not to repeat the same reference twice.

Table 2 (approx. line 288 onwards): I am not convinced that this needs to be in the main text. I would suggest moving this to the appendix.

- Table 2 (approx. line 288 onwards): We accept the suggestion to move Table 2 to the Appendix.

Figure 8: There is a lot going on in this figure which makes difficult to get an eye in and the textbox in plot a overlaps with the data presented. I'm not completely sure how best to improve it, but removing gridlines, text box edges and repeated x-axis labels might help.

- Figure 8: To increase the readability of Figure 8, we suggest making the gridlines more transparent and removing the repetitive x-axis labels. We find that this solution improves the clarity of the figure significantly, while some consistency between this and other figures of the manuscript remains.

Line 334: Does density act as a greater control of SWEmax than maximum snowpack height? Could you add something about relative snow depth differences between sites to this paragraph?

- Line 334: The density difference between low and high SWE sites is less important than the HS difference between these sites and can for this discussion be ignored. Still, the original phrasing of sentence L334 could have been clearer. We propose to rewrite the sentence, removing the word 'dense' and instead referring to the sites with higher average $SWE_{max}$ values.

  'The snow cover in areas with higher average $SWE_{max}$ values is less sensitive to variability in solid precipitation and wind because the relative changes resulting from variability in these parameters are small at these sites.'

Line 363: How long is "relatively short"? Please quantify the length of the timeseries.

- Line 363: We accept the suggestion to quantify the length of the time series. Changed sentence: 'It should be noted that the QIN data should not be overinterpreted, as the weather station is not located ideally and the time series is only four years long, with a gap of one year.'

Line 371: What is the GEMP? Please define this acronym

- Line 371: We changed 'GEMP' to Greenland Ecosystem Monitoring Project. Introducing the abbreviation is not needed as we only refer to this project once.

Line 421: Other studies suggest that changes in the timing of snow-off will not lead to changes in the length of the growing season (e.g., Kelsey et al. 2021; Semenchuk et al, 2016; Starr et al. 2000). Please justify.

- Line 421: We recognize that the link between snow cover duration and plant productivity is less clear than we originally posed. We suggest to nuance our statement and reflect on the complexity of these interactions in the following way: "Shorter $SC_{duration}$ can in some areas with enough soil moisture increase plant productivity and carbon capture in areas with enough soil moisture (Callaghan et al., 2012), however, temperature has been indicated to be a more important driver for the length of the growing season (Kelsey et al., 2021). These vegetation-snow interactions remain complex, species species-dependent and are still not fully understood."

Line 432: Do the limitations need to be a separate subsection when this is only 2 sentences long?

- Line 432: We do believe the limitations should be a separate subsection, independent of the length of the text. This is because including the limitations in another subsection would mean that that subsection heading no longer adequately covers its content.

Line 466 : This is a very unusual place to put a summary. If you wish to include this, consider moving it to the beginning of the article.

- Line 466: The summary should indeed appear only on the online page from Cryosphere and not in the manuscript itself. We removed the summary at the end of the manuscript.

Line 615: What does the grey dashed line represent in Figure A1?

- Line 615: We removed the grey dashed lines in Figure A1 and Figure 5, as they did not have any meaning in the current version of the figures.

Line 623: The caption says "all sites" but only 5 of the 9 sites included are shown in Figure A2.

- Line 623: This is because not all sites have racmo2.3p2 data available. We suggest changing the sentence in the following way so that this becomes more clear:

'Daily SWE$\Delta$snow values are plotted against SWERACMO values for each location with available RACMO2.3p2 data.'

Appendix B: These tables are not referred to in the main text, and I am not convinced that they are needed.

- Appendix B: We would prefer to keep the information in Table 2 and the tables in the Appendix, they are referred to in L374. The information in these tables provides a more detailed report on the spatio-temporal variability of the different snow indicators and could be useful for other researchers who want to compare similar values from different regions.